# Membrane Fouling Mechanism of HTR-PVDF and HMR-PVDF Hollow Fiber Membranes in MBR System

**Kaikai Chen [1,\*], Wei Zhao [2], Changfa Xiao [1], Hui Zhu [1] and Qiming Wang [1]**

1    School of Textiles and Fashion, Shanghai University of Engineering Science, Shanghai 201620, China
2    School of Materials Science and Engineering, Tiangong University, Tianjin 300387, China
\*    Correspondence: chenkaikai@sues.edu.cn

**Abstract:** Membrane fouling has attracted a lot of attention in the membrane separation field. Herein, we selected the homogeneous-reinforced polyvinylidene fluoride (HMR-PVDF) and heterogeneous-reinforced polyvinylidene fluoride (HTR-PVDF) hollow fiber membranes to investigate the fouling mechanism of membranes in membrane bioreactor (MBR) systems. The filtration models, membrane adsorption experiment, and membrane resistance distribution after a long or short time operation were assessed to compare their antifouling properties in order to verify the optimal membrane. The outer surface, shown by an SEM observation of the HMR-PVDF and HTR-PVDF membranes, was rough and smooth, respectively. Moreover, the HMR-PVDF membranes had a higher adsorption capacity than the HTR-PVDF membranes when an equilibrium state was almost 2.81 times that of the original membrane resistance. A cleaning method (mainly physical and chemical) was utilized to illustrate the operational stability of the membranes. In summary, the HMR-PVDF hollow fiber membrane presented better antifouling properties than the HTR-PVDF membranes, which was conducive to industrial implementation.

**Keywords:** reinforced hollow fiber membrane; polyvinylidene fluoride; membrane bioreactor; fouling mechanism; antifouling property



## 1. Introduction

It is well-known that membrane bioreactor (MBR) technology is a combination of the conventional activated sludge (ASP) and membrane separation processes, which have been applied to wastewater treatment. As is known, industrial dairy waste contains a high concentration of organic material, such as proteins, which have high levels of chemical oxygen demand (COD), total Kjeldahl nitrogen (TKN), and high concentrations of suspended solids (SS). Through membrane filtration, wastewater could be partially reused or followed by deep filtration [1–3]. Especially in the MBR system, the hollow fiber membrane was prone to be affected by long-term water flow pulsation or disturbance, which was easily damaged and destroyed by high-pressure hydraulic cleaning processes, high-speed water flow disturbances, or even aeration processes. Up to now, the enhanced mechanical properties of hollow fiber membranes could meet the actual requirement by the reinforced method [4–6]. In general, the reinforced hollow fiber membranes were usually constructed by introducing a braid tube. Moreover, the homogenous reinforced (HMR) and heterogeneous-reinforced (HTR) hollow fiber membranes were put forward through the same or different separation and supporting layers [7–9].

Nowadays, polyvinylidene fluoride (PVDF) hollow fiber membranes are widely used in MBR systems. Many works have reported on the improvements in antifouling for PVDF hollow fiber membranes. For example, Hikita et al. [10] studied amphiphilic graft copolymers as coatings on commercially available PVDF membranes. Although the composite membranes possessed good antifouling properties, the separation (coating) layer was easily peeled off. Liu et al. [11] also designed antifouling PVDF membranes by blending

TiO$_2$. It could be found that this method plays an important in the antifouling of the outer surface. However, the large pore structure was generated with pore blocking, which might develop permanent fouling on PVDF hollow fiber membranes. So far, reinforced hollow fiber membranes, including heterogeneous reinforced and homogeneous reinforced, were suitable for overcoming the limitations of poor tensile strength. In the beginning, the heterogeneous reinforced membrane was first proposed by coating the polymer solutions on the outer surface of the high-strength hollow tubular braid. However, thermodynamic incompatibility between the reinforced fiber and porous membrane caused the separation layer to be easily peeled from the tubular braid. Compared with the aforementioned heterogeneous reinforced method, the advantage of the homogeneous reinforced method that contained the same materials in the separation layer and the supporting layer is that it exhibited thermodynamic compatibility, having good interfacial bonding strength. The results indicated the favorable interfacial bonding between the separation layer and the high-strength hollow tubular braid, avoiding peeling. However, the favorable interfacial bonding could produce a dense interface, which also affects the permeability [12–14].

Based on previous work, the fouling mechanism of the membrane separation process, including reversible and irreversible fouling or the formation of cake and gel layers, has also been widely researched [15,16]. On the one hand, an understanding of the fouling mechanism could help optimize the structure and properties of the membranes. On the other hand, it is important to prolong the service life of membranes during the MBR process. For instance, Du [17] found that one of the vital factors was the hydrophilic and hydrophobic properties of the membrane, which could result in faster irreversible fouling with more hydrophobicity. Consequently, the fouling mechanism and antifouling properties of the reinforced PVDF hollow fiber membrane will exhibit promising research significance for a wide range of membrane applications in MBR systems.

In this work, PVDF and polyacrylonitrile (PAN) braid tubes were selected to be the supporting layer for HMR-PVDF and HTR-PVDF hollow fiber membranes to study their fouling properties. These two kinds of hollow fiber membranes were prepared by using concentric spinning technology, as in our previous work [18,19]. The fouling mechanism and the characterization of the membrane properties could also be summarized. The purpose of the study is to provide a scientific basis for improving and further applying reinforced hollow fiber membranes in MBR systems.

## 2. Experimental

### 2.1. Materials

The chemicals used in this study were supplied by Tianjin Kermel Chemical Reagent Co., Ltd. (Tianjin, China). Bovine serum albumin (BSA, Mw = 68,000) was purchased from Sinopharm Chemical Reagent Co., Ltd. (Shanghai, China).

### 2.2. Membrane Characteristic

An optical contact angle meter (DSA130, KRUSS) and a rheometer (HAAKE MARS, Thermo Fisher Scientific, Dreieich, Germany) were utilized to measure the water contact angle of the membrane and the viscosity of the casting solutions. The maximum pore size and mean pore size distribution were tested using bubble point test equipment (3H-2000PB, Beishide, Beijing, China). The porosity of the hollow fiber membrane was regarded as the volume of pores in the total volume. In general, the gravimetric method was usually utilized to calculate it. Firstly, deionized water was selected to soak the membrane sample for 24 h. Then a blowing system was employed to remove the water in the pores of the membrane. Then, a paper filter was utilized to remove the water attached to the outer membrane surface. Finally, the weight of the wet membrane and dry membrane was tested after drying in an electric blast drying oven for 12 h at a temperature of 40 °C.

### 2.3. Feed Water Characteristic

The feed water was prepared by mimicking industrial wastewater. The water quality is presented in Table 1.

**Table 1.** The composition of feed water.

| Components | Concentration (mg/L) | Components | Concentration (mg/L) | Components | Concentration (mg/L) |
|---|---|---|---|---|---|
| Glucose | 300.00 | Carbamide | 30.00 | Ferric Chloride | 0.25 |
| Sodium Bicarbonate | 30.00 | Magnesium Sulfate | 12.00 | Calcium Chloride | 6.00 |
| Sodium Dihydrogen Phosphate | 12.75 | Manganese Sulfate | 6.00 | Sodium Chloride | 150.00 |

### 2.4. Filtration Models

In the filtration process, both membrane pore blocking and cake layer formation on the membranes were usually the predominant fouling component. The filtration models contributed to predicting membrane fouling. The filtration process could be described by the standard pore-blocking filtration model and cake filtration law [20]. The two models could be calculated by:

Pore-blocking resistance:

$$\frac{K_p T}{2} = \frac{T}{V} - \frac{1}{Q_0} \tag{1}$$

Cake resistance:

$$\frac{K_c V}{2} = \frac{T}{V} - \frac{1}{Q_0} \tag{2}$$

where $K_p$, and $K_c$ are the parameters relating to pore-blocking resistance and cake formation resistance, respectively; $T$ is the operation time; $V$ is the volume of permeation.

Under a constant pressure of 0.1 MPa, the volume of the water was recorded every 5 min, and the dominant position of the two laws can be judged by the charts of $T \sim \frac{T}{V}$, $V \sim \frac{T}{V}$.

### 2.5. Membrane Adsorption

The cross-flow filtration of membrane could be described by Darcy's law [21] as:

$$J_{0a} = \frac{\Delta P}{\mu_0 R_{\mathrm{ma}}} \tag{3}$$

where $J_{0a}$ (L·m$^{-2}$·h$^{-1}$) is the pure water flux; $\Delta P$ (Pa), the trans-membrane pressure (TMP), $\mu_0$ (Pa·s) is the pure water viscosity, and $R_{ma}$ (m$^{-1}$) is the membrane resistance.

$$J_{1a} = \frac{\Delta P}{\mu_1 R_{ta}} \tag{4}$$

where $J_{1a}$ (L·m$^{-2}$·h$^{-1}$) is the mixed liquor flux; $\Delta P$ (Pa) the trans-membrane pressure (TMP), $\mu_1$ (Pa·s) is the mixed water viscosity, and $R_{ta}$ (m$^{-1}$) is the total membrane resistance after it is polluted.

$$R_{ca} = R_{ma} \frac{\Delta P}{\mu_0 J_{2a}} \tag{5}$$

$$R_a = R_{ta} - R_{ca} - R_{ma} \tag{6}$$

where $R_{ca}$ is the resistance of cake layer; $R_a$ (m$^{-1}$) is the membrane adsorption resistance.

The MBR system is shown in Figure S1. The HMR-PVDF and HTR-PVDF membranes were submerged in the biochemical pond for 30 days to study the adsorption process in

a static method. Then the surface of the membranes was cleaned slightly with absorbent cotton to remove the reversible effect of fouling. The membrane resistance was calculated and used to identify the fouling mechanism due to the adsorption phenomenon of the two membranes. The purpose was to study the adsorption phenomenon of the two membranes induced by the static procedure.

*2.6. Fouling Mechanism*

With sustainable filtration, the increase in membrane resistance led to flux decline, and the fouling mechanism usually led to the formation of pore blocking, concentration polarization, gel layer, and cake layer. Moreover, the degree of membrane fouling expressed by the resistance was calculated in the following model:

$$J_0 = \frac{\Delta P}{\mu_0 R_m} \tag{7}$$

where $J_0$ (L·m$^{-2}$·h$^{-1}$) is the pure water flux; $\Delta P$ (Pa) is the trans-membrane pressure (TMP); $R_m$ (m$^{-1}$) is the clean membrane resistance.

$$J_1 = \frac{\Delta P}{\mu_1 R_t} \tag{8}$$

$$R_t = R_m + R_i + R_g + R_c + R_{cp} \tag{9}$$

where $J_0$ (L·m$^{-2}$·h$^{-1}$) is the mixed liquor flux. $R_t$ (m$^{-1}$) is the total filtration resistance; $R_i$ (m$^{-1}$) is the irreversible resistance; $R_g$ (m$^{-1}$) is the gel layer resistance; $R_c$ (m$^{-1}$) is the cake layer resistance; and $R_{cp}$ (m$^{-1}$) is the resistance due to concentration polarization.

The concentration polarization resistance ($R_{cp}$), the cake layer resistance ($R_c$), the gel layer resistance ($R_g$), and the pore blocking resistance ($R_i$) were calculated following the model below:

$$R_{cp} = R_m - \frac{\Delta P}{\mu_0 J_2} \tag{10}$$

$$R_c = R_m - R_{cp} - \frac{\Delta P}{\mu_0 J_3} \tag{11}$$

$$R_g = R_m - R_{cp} - R_c - \frac{\Delta P}{\mu_0 J_4} \tag{12}$$

$$R_i = R_m - R_{cp} - R_c - R_g \tag{13}$$

where $J_2$ (L·m$^{-2}$·h$^{-1}$) is the pure water flux; $J_3$ (L·m$^{-2}$·h$^{-1}$) is the pure water flux after physical cleaning; $J_4$ (L·m$^{-2}$·h$^{-1}$) is the pure water flux after chemical cleaning.

The HMR-PVDF and HTR-PVDF membranes were submerged in the MBR system for 45 d. After a 12 h operation, the degree of membrane fouling can be investigated by $R_t$. Then, then the membranes were cleaned with absorbent cotton, and the $R_c$, $R_{cp}$, and $R_i$ were calculated separately to study the membrane fouling degree of the initial time. After 45 d in MBR, the degree of membrane fouling was calculated using the resistance following the model above. In MBR, the common chemical cleaning agents were inorganic or organic acids and sodium hypochlorite. The acid was used to remove the inorganic pollutants, while sodium hypochlorite (NaClO) was used for organic pollutants. In this process, the membranes were also cleaned with absorbent cotton, and then chemical cleaning: one method was to immerse the reinforced PVDF hollow fiber membranes in 1% citric acid followed by 0.5% sodium hypochlorite solution for about 24 h, and the other was to reverse the immersion sequence of the two chemical agents.

### 2.7. Morphology Observation

The morphology of the original, polluted, and cleaned membranes could be observed using a scanning electron microscope (SEM Pharos G1 and FESEM S4800). The samples should be freeze-dried at first, then sputtered with gold and recorded through SEM.

## 3. Results and Discussions

### 3.1. Filtration Models

As is well-known, sludge cake deposition and pore blocking are the main means of representation for membrane fouling in MBR systems [22]. In the actual experiment, the formation of the cake layer and pore blocking were formed at the same time due to the mixture consisting of colloids, solutions, and a mass of particles. As is shown in Figure 1, the fouling models of the HMR-PVDF and HTR-PVDF hollow fiber membranes were established by fitting the experimental data. It could be seen that the formation of the cake layer was formed firstly and was mostly occupied during the filtration process. In addition, these two kinds of hollow fiber membranes possessed ideal antifouling properties and avoided pore blocking effectively, with $V$ linearly associated with $\frac{T}{V}$. Consequently, the research focus of HMR-PVDF and HTR-PVDF hollow fiber membranes should be mainly on the physical and chemical cleaning of the cake or gel layer, which is discussed below.

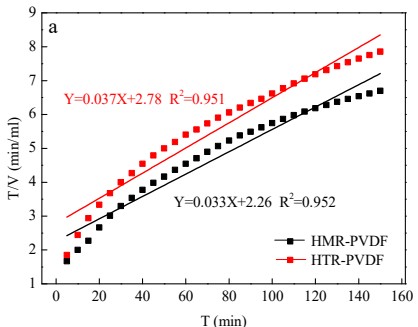
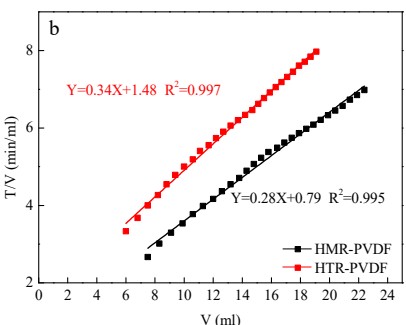

**Figure 1.** Determination of fouling mechanisms from model fitting to experimental data (dotted line indicates the fitted straight line). (**a**) Fitting of pore blocking model, (**b**) Fitting of cake layer model.

### 3.2. Membrane Adsorption

Surface adsorption played an important role in the membrane fouling process [23]. The HMR-PVDF and HTR-PVDF hollow fiber membranes were immersed in the MBR system (MLSS = 9600 mg/L), respectively. The changes in membrane resistances during the adsorption process are shown in Figure S2. The total membrane resistance of the HMR-PVDF membrane induced by the static procedure was very small initially and then increased rapidly in the following time. After immersion for about 600 h, the total membrane resistance that reached an equilibrium state was almost 2.85 times that of the original membrane resistance. While the HTR-PVDF membrane resistance increased rapidly in the initial time and then this increase was small. After immersion for about 110 h, the HTR-PVDF membrane resistance that arrived at an equilibrium state was almost 2.81 times that of the original membrane resistance. Under the same condition, the HMR-PVDF membranes had a higher adsorption capacity compared with the HTR-PVDF membranes. They both exhibited equilibrium states. In a test using direct observation of the membrane operating in a cross-flow system, the flocculent materials were visually observed to deposit temporarily on the membrane. In a word, the increase in the HTR-PVDF membrane resistances was mainly due to rapid adsorption, while the HMR-PVDF membrane was delayed in reaching an equilibrium state, that is to say, the HMR-PVDF hollow fiber membranes might exhibit better antifouling than the HTR-PVDF hollow fiber membranes.

In order to verify the antifouling properties of the membranes, the values for membrane resistance distribution due to adsorption are shown in Figure 2. As mentioned above,

the increase in the HTR-PVDF membrane resistances was mainly due to rapid adsorption. The membrane adsorption resistance ($R_{ad}$) of the HTR-PVDF membrane never exceeded 41.47%, while the HMR-PVDF membrane is only 20.17%. As shown in Table 1, the HMR-PVDF membranes were more hydrophilic than the HTR-PVDF membranes. Considering the hydrophobic interactions [24], the HTR-PVDF hollow fiber membranes exhibited good adsorption and deposition of pollutants on the membrane surface, which could form a denser adsorption layer than the HMR-PVDF membranes, the HTR-PVDF hollow fiber membranes. As observed in Figure 3A, the HTR-PVDF membranes were smooth and inclined to achieve an equilibrium state and formed a thick layer during the static adsorption process. Moreover, the outer surface SEM morphologies of the HMR-PVDF membranes are shown in Figure S3. Figure S3a shows that the HMR-PVDF membranes were rough, with many "valleys" on the surfaces. Particles were preferentially prevented in the "valleys" due to the size of the "valleys" being less than the size of the particles, which delayed the equilibrium state more than the smooth membranes [25]. Therefore, the HTR-PVDF membranes tend to form a thick cake layer owing to many particles or fluctuant deposits on the membrane surface, which might result in the formation of a gel layer associated with hydrophobic and chemical bond interactions [26]. As for the difference between the surface adsorption of the two kinds of membranes, rough membranes would probably possess a much lower flux resistance than the dense film formed on the smooth membranes. Figure 3B shows the polluted surface of the HMR-PVDF membrane. The membrane surface was expected to be mostly covered by adsorption, promoting the attachment of microbial cells or fluctuant. Furthermore, the polluted outer surface was very rough, and the microbial cells were visible on the surface. Figure S3b shows the polluted surface of the HTR-PVDF membrane. The membranes were covered with a denser cake layer, and the outer surface looked smooth. The separation layers of the HMR-PVDF and HTR-PVDF hollow fiber membranes had different adsorption capacities due to the different hydrophobicity and roughness. In the static adsorption procedure, irreversible adsorption fouling was observed for the cross-flow operation. In short, the HMR-PVDF hollow fiber membranes exhibited good antifouling due to physical absorption, as aforementioned. It was consistent with the results of the filtration models.

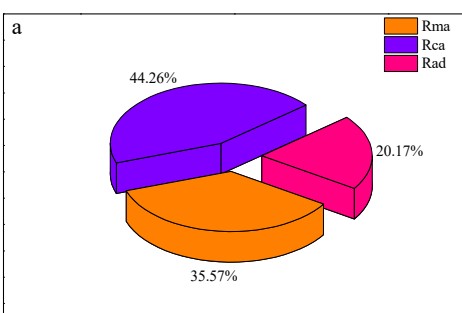 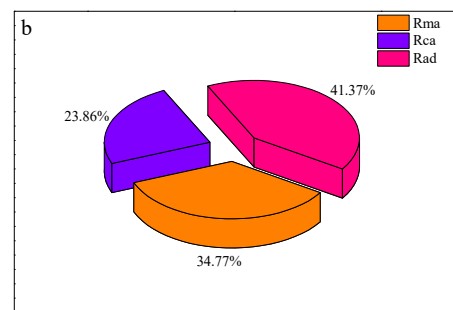

**Figure 2.** The membrane resistance distribution of adsorption (**a**) HMR-PVDF (**b**) HTR-PVDF.

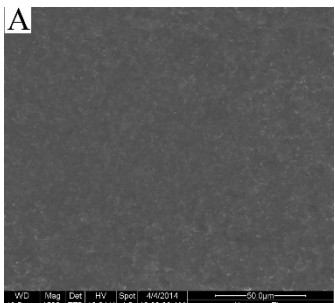 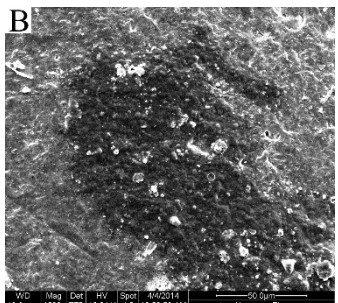

**Figure 3.** Outer surface SEM morphology of the HMR-PVDF membrane (**A**) initial, (**B**) adsorption.

*3.3. Fouling Mechanism*

The fouling mechanisms of the membranes were very complex. As discussed above, the HMR-PVDF hollow fiber membranes exhibited better antifouling than the HTR-PVDF hollow fiber membranes. In general, the size effects were usually considered during the fouling process. If the sizes of the foulants are comparable with the membrane pores (i.e., colloids) or smaller than the membrane pores (i.e., solutes), they may be responsible for adsorption on the pore wall or pore plugging. However, if the foulants (i.e., sludge flocs and colloids) are much larger than the membrane pores, they tend to form a cake layer on the membrane surface [27]. A cake layer will develop with reversible fouling, while pore blocking and membrane compaction tend to develop into irreversible fouling [28–30]. In this work, we found that the membrane fouling mechanism of the PVDF hollow fiber membranes might be related to the adsorption of solutes or colloids within/on membranes, the precipitation of sparingly-soluble macromolecular polymeric and inorganic, and the accumulation of retained solids on the membrane, according to the adsorption experiment. In brief, the HMR-PVDF hollow fiber membranes with the cake layer could be removed by physical cleaning, while the HTR-PVDF hollow fiber membranes with the gel layer need to be removed by chemical cleaning during the actual application. Furthermore, the membrane resistance distribution after both long and short operation was tested to verify the results.

### 3.3.1. The Membrane Resistance Distribution after Short-Time Operation

The properties of the HMR-PVDF and HTR-PVDF membranes are summarized in Table 2. These two kinds of membranes exhibited a similar average pore size and porosity due to the same double diffusion behavior during the membrane preparation process. The HMR-PVDF hollow fiber membrane with a higher protein rejection rate and pure water flux exhibited more hydrophobicity than that of the HTR-PVDF hollow fiber membrane.

**Table 2.** Properties of the membranes used in this work.

| Items | HTR-PVDF | HMR-PVDF |
|---|---|---|
| Average Pore Size (μm) | $0.58 \pm 0.01$ | $0.53 \pm 0.02$ |
| Porosity (%) | $45.67 \pm 0.2$ | $50.19 \pm 0.35$ |
| Contact Angle (°) | $74.50 \pm 0.5$ | $72.00 \pm 0.6$ |
| Protein Rejection Rate (%) | $96.03 \pm 1.1$ | $97.88 \pm 0.9$ |
| Pure Water Flux ($L \cdot m^{-2} \cdot h^{-1}$) | $269.34 \pm 2.2$ | $293.82 \pm 2.5$ |

Figure 4 shows the membrane resistance distribution of the HTR-PVDF and HMR-PVDF membrane after operation for 12 h in MBR, respectively. As shown in Figure 4a, the cake layer, which accounted for 66.42% of the total membrane resistance, was considered to be the major contributor to membranes' permeation decay, while the HMR-PVDF membranes took 82.55% of the total resistance, as shown in Figure 4b. However, the cake layer occupied only 1.14%, which had not fully developed in 12 h. The HMR-PVDF membranes possessed better hydrophilicity than the HTR-PVDF membranes and more easily developed a hydro-layer close to the membrane surface, which hindered colloids or fluctuant from depositing on the membrane surface to form a denser deposit layer in the initial time. Table 3 shows the flux changes of the HTR-PVDF and HMR-PVDF membrane interval after 12 h in MBR. It shows that the HMR-PVDF membrane possessed a lower flux decay than that of the HTR-PVDF membrane. During the filtration process, concentration polarization, which was a phenomenon due to the solute accumulating on the membrane, was the main factor promoting and generating membrane fouling. Figure 4b indicates that the fouling layer of the HMR-PVDF membrane is almost negligible. The influence of concentration polarization disappeared when the filtration process terminated. The concentration polarization phenomena were also responsible for the declining permeation flux. In the MBR system, the control of concentration polarization ultimately determined the successful suppression of fouling [31]. The $R_i$ values of the HMR-PVDF membrane

were zero, indicating that the pollute layer might be completely removed by physical cleaning with absorbent cotton. Consequently, the HMR-PVDF hollow fiber membranes with fouling could be removed by physical cleaning.

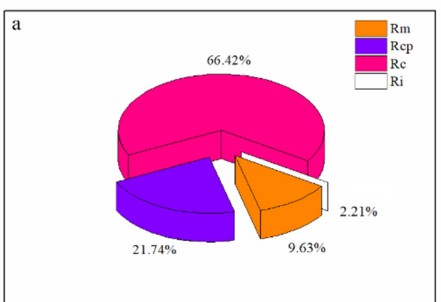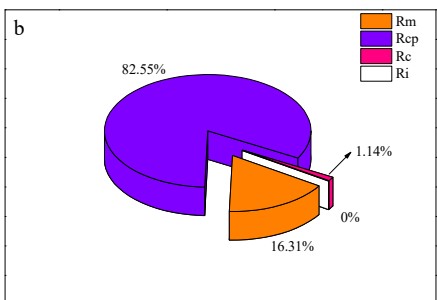

**Figure 4.** The membrane resistance distribution after 12 h in MBR (**a**) HTR-PVDF, (**b**) HMR-PVDF.

**Table 3.** The flux changes of the HTR-PVDF and HMR-PVDF membrane after 12 h in MBR.

| Items | HTR-PVDF | HMR-PVDF |
|---|---|---|
| $J_0$ (L·m$^{-2}$·h$^{-1}$) | 218.38 | 251.59 |
| $J_1$ (L·m$^{-2}$·h$^{-1}$) | 213.65 | 221.66 |
| $J_2$ (L·m$^{-2}$·h$^{-1}$) | 177.00 | 181.50 |
| $J_3$ (L·m$^{-2}$·h$^{-1}$) | 155.25 | 171.56 |

### 3.3.2. The Membrane Resistance Distribution after Long-Time Operation

The cake and gel layers were compressed in the long-term operation. During the long-term operation of MBR, foulants blocked the pores and formed a strongly-attached fouling layer. At the same time, some foulants might progressively deposit onto the membranes or into the membrane pores. The membrane resistance distributions after 45 d in MBR are presented in Figure 5. For the HTR-PVDF and HMR-PVDF hollow fiber membranes, their membrane resistance distribution under long-term operation exhibited a significant difference. Figure 5a shows that the concentration polarization, which occupied about 50% of the total membrane resistance, was the main reason for the permeation flux decay for the HMR-PVDF membrane. A deposited cake and gel layer compression increases the difficulty of removing the foulants by physical cleaning. For the HTR-PVDF membrane, Figure 5b shows that the gel layer, which took about 42% of the total resistance, was the dominant contributor to membrane fouling. The adsorption phenomenon of the HTR-PVDF membranes discussed above indicated that it was inclined to adsorb solutes or colloids as well and tended to develop a dense gel layer. Compared with the membrane resistance distribution in the short term, the influence concentration polarization phenomena, whose proportion had decreased to 28%, was weak. In the flitting process, the deposition resistance of the HMR-PVDF membrane was about 40% that of the total resistance, and the cake layer was about 25%. The outer surface SEM morphology changes of the HMR-PVDF membrane are shown in Figure S4. Figure S4a indicates that the polluted HMR-PVDF membrane, whose surface was rough, developed a loose cake layer during long-term operation. However, the proportion of the HTR-PVDF membrane's deposition resistance was higher, which reached about 65%, and the gel layer was about 42%. The outer surface SEM morphology changes of the HTR-PVDF membrane are shown in Figure S5. Figure S5a indicates that the polluted HTR-PVDF membrane, whose surface was smooth, developed a denser fouling layer. The roughness of the membrane surface played a key role in particle adhesion as well. It indicated that a loose fouling layer was produced in the rough membranes (HMR-PVDF), resulting in a low flow resistance (per unit thickness of foulant) rather than the dense fouling layers (HTR-PVDF). Citric acid buffering aims to remove the minerals from the fouling layers, and sodium hypochlorite (NaClO) was found to remove the organics well. The cleaning sequence was known to affect the degree of permeability

recovery. Different choices of cleaning sequences achieved different effects, which could be explained by the charge on the membrane surface and the type of fouling. It appeared that organic matter, alkali, acid, etc., resulted in different kinds of fouling. The flux changes of the HTR-PVDF and HMR-PVDF membrane after 45 d in MBR are shown in Table 4. It indicated that the most effective cleaning sequence was critic acid followed by sodium hypochlorite. The flux of the HTR-PVDF membrane almost recovered to 96% after the two kinds of membranes were immersed in 1% citric acid for 20 min, followed by 0.5% sodium hypochlorite for 20 min, whilst the HMR-PVDF membrane achieved 99% after immersion in the two chemical agents for 60 min, respectively. Figures S4b and S5b show that after being cleaned by sodium hypochlorite-citric acid, the fouling layer and microbial cells could be clearly observed on the membrane surface. However, Figures S4c and S5c indicate that with citric acid–sodium hypochlorite cleaning, the fouling matters were almost removed, and nothing could be seen on the membrane surface. Hence, the HTR-PVDF and HMR-PVDF hollow fiber membranes, with a predominantly hydrophobic nature, favored organic matter. Under the same conditions of chemical cleaning, the HMR-PVDF membrane could achieve an ideal effect. These rough membranes with a loose surface fouling layer illustrated low flow resistance per unit thickness of foulant compared with the dense fouling layers observed on the smooth-surfaced membranes. Therefore, the HMR-PVDF hollow fiber membrane with a rough surface and a loose fouling layer could be cleaned easily by physical and chemical cleaning.

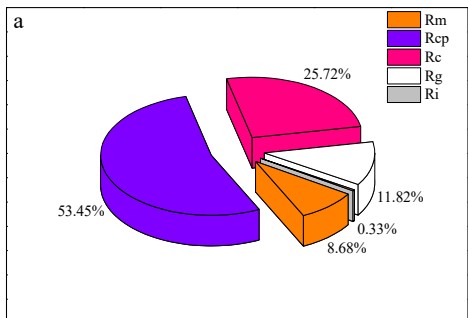 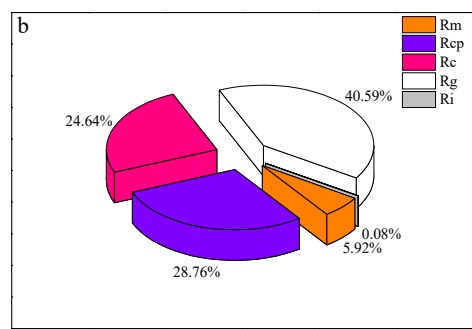

**Figure 5.** The membrane resistance distribution after 45 d in MBR. (**a**) Immersed the HTR-PVDF membrane in critic acid (20 min) + NaClO (20 min). (**b**) Immersed the HMR-PVDF membrane in critic acid (60 min) + NaClO (60 min).

**Table 4.** The flux changes of the HTR-PVDF and HMR-PVDF membrane after 45 d in MBR.

| Items | HTR-PVDF [a] | HTR-PVDF [b] | HMR-PVDF [a] | HMR-PVDF [b] |
|---|---|---|---|---|
| $J_0$ (L·m$^{-2}$·h$^{-1}$) | 218.38 | 218.38 | 246.67 | 251.59 |
| $J_1$ (L·m$^{-2}$·h$^{-1}$) | 16.68 | 19.43 | 12.85 | 10.83 |
| $J_2$ (L·m$^{-2}$·h$^{-1}$) | 40.72 | 42.08 | 20.50 | 17.21 |
| $J_3$ (L·m$^{-2}$·h$^{-1}$) | 90.99 | 91.04 | 31.34 | 23.89 |
| $J_4$ (L·m$^{-2}$·h$^{-1}$) | 210.42 | 195.63 | 243.20 | 168.79 |

HTR-PVDF [a] and HTR-PVDF [b]: immersed the HTR-PVDF membrane in critic acid (20 min) + NaClO (20 min) and NaClO (20 min) + critic acid (20 min), respectively; HMR-PVDF [a] and HMR-PVDF [b]: immersed the HMR-PVDF membrane in critic acid (60 min) + NaClO (60 min) and NaClO (60 min) + critic acid (60 min), respectively.

The particle size distribution of the activated sludge in MBR is shown in Figure 6. The minimum particle size was bigger than the maximum pore size of the HTR-PVDF and HMR-PVDF membranes, and thus the sludge fluctuant would hardly plug into the pores. It coincided with the filtration discussed above that the gel layer took the dominant occupation, which led to the development of irremovable fouling on the membranes under long-term operation. In the filtration process, the irremovable fouling resistance of the membranes was caused by pore blocking and the irreversible adsorption of foulants onto the membrane pore wall or surface. The cross-sectional SEM morphologies of the hollow

fiber membranes are shown in Figure S6. It shows that no particles or colloids were attached to the membrane pores and resulting in pore plugging. As predicted in the filtration models, that cake formation fouling almost dominated the filtration process except for the pore blocking model. However, HTR-PVDF and HMR-PVDF membrane resistance was mainly deposit resistant and concentration polarization resistant. Figure 6 and Figure S6 indicate that the reason for the irremovable fouling was the irreversible adsorption of foulants onto the membrane surface. As for the HTR-PVDF membranes, the adsorbed foulants were more difficult to remove than on the HMR-PVDF membranes. Because the foulants on the surface of the HTR-PVDF membrane might form a dense gel layer. In brief, HMR-PVDF possessed great antifouling properties and mechanical strength in MBR operation. Figure 5 shows that the deposited layer of the HMR-PVDF membranes produced a cake layer, while the HTR-PVDF membrane had a gel layer after long-term operation in the MBR, which coincided with the conclusion discussed in the adsorption phenomenon.

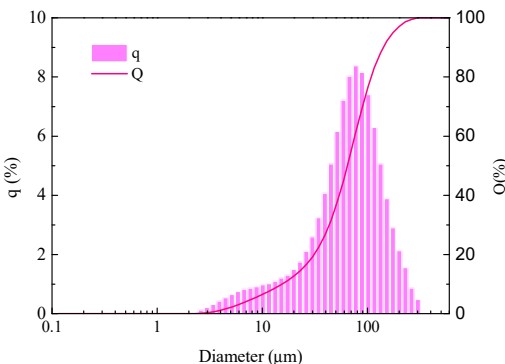

**Figure 6.** Particle size distribution of activate sludge in MBR.

## 4. Conclusions

In this work, the HMR-PVDF and HTR-PVDF hollow fiber membranes were both preferably fitting the formation model of the cake layer. Under the same condition, the HMR-PVDF membranes had a higher adsorption capacity than the HTR-PVDF membranes, which might be illustrated by the delayed pore blocking. The membrane resistance distribution of these two kinds of membranes in the long-term operation showed great differences in the short term. The flux changes of the HTR-PVDF and HMR-PVDF membrane after 45 d in the MBR system exhibited that the recovery flux of HTR-PVDF was almost 96%, while the HMR-PVDF membrane achieved the 99%. Moreover, the SEM morphologies indicated that the rough surface of the HMR-PVDF membrane performed better against fouling than the HTR-PVDF membrane due to the delayed formation of the gel layer. Furthermore, the HMR-PVDF hollow fiber membrane, after a long-term operation, could be cleaned by physical cleaning, which could save costs in industrialization processes. In summary, the HMR-PVDF hollow fiber membrane with good antifouling properties verified by the fouling model and experiment shows a broad application prospect in the membrane separation field.

**Supplementary Materials:** The following supporting information can be downloaded at: https://www.mdpi.com/article/10.3390/w14162576/s1, Figure S1: The installation diagram of MBR system; Figure S2: The change of membrane resistance during adsorption process; Figure S3: Outer surface SEM morphology of the HTR-PVDF membrane (A) initial, (B) adsorption; Figure S4: Outer surface SEM morphology of the HMR-PVDF membrane(a) initial, (b) cleaned with NaClO (20 min) + critic acid (20 min), (c) cleaned with critic acid (20 min) + NaClO (20 min); Figure S5: Outer surface SEM morphology of the HTR-PVDF membrane(a) initial, (b) cleaned with NaClO (60 min) + critic acid (60 min), (c) cleaned with critic acid (60 min) + NaClO (60 min); Figure S6: Cross-sectional SEM morphology of hollow fiber membranes (a) HTR-PVDF (b) HMR-PVDF.

**Author Contributions:** K.C. and W.Z. carried out the experiment. K.C. wrote the manuscript with support from W.Z., H.Z. and Q.W. fabricated the SEM sample. K.C. and C.X. helped supervise the project. All authors have read and agreed to the published version of the manuscript.

**Funding:** This work was supported by the National Natural Science Foundation of China (52103035 and 52173038).

**Informed Consent Statement:** Not applicable.

**Conflicts of Interest:** The authors declare no competing financial interest.

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
