# Peer review of "Membrane Fouling Mechanism of HTR-PVDF and HMR-PVDF Hollow Fiber Membranes in MBR System"

_water, doi:10.3390/w14162576_

Round 1
Reviewer 1 Report
Point 1: The English of the text should be checked throughly.
Point 2: The novelty of manuscript should be highlighted more.
Point 3: Preparation methods of HMR-PVDF and HTR-PVDF hollow fiber membranes should be shown briefly.
Point 4:Line 61-67 should be moved to “Results and discussion” part. The corresponding determining methods of items in Table 1 should be added in Experimental part. Chemicals used in this study should be presented in this part as well.
Point 5: Data in Tables were measured for only once? If not, please identify determination times and the data in the table should be indicated as average value ± standard deviation. Unit in table 1 should be double-checked as well.
Point 6: Line 198-201 should be rewritten.
Point 7: Format of Line 203 should be double-checked.
Point 8: Roughness data (Ra, Rq and Rmax) of HTR-PVDF and HMR-PVDF membranes should be shown in this paper.
Point 9: Format in this paper should be double-checked, such as “Table”/”Tab.”
Author Response
Reviewer: #1
Technical Aspects:
- Comment: The English of the text should be checked throughly
Response: Considering the Reviewer’s suggestion, we have revised the full paper carefully and tried to reframe the sentences with proper word and tenses. For example, the first sentence in Abstract was improved as " Membrane fouling has attracted a lot of attention in membrane separation field. ". In addition, the whole revised manuscript has been thoroughly edited by a native English speaker from an editing company.
- Comment: The novelty of manuscript should be highlighted more
Response: As the reviewer suggested, we have revised the highlights so as to make it clear. The revised details were as following:
- The HMR-PVDF and HTR-PVDF hollow fiber membrane were prepared to verify the Membrane fouling mechanism.
- The HMR-PVDF membranes exhibited better adsorption capacity compared with the HTR-PVDF membranes.
- The good anti-fouling property of HTR-PVDF membrane was due to deposit colloidal particles and a loose cake layer.
- Comment: Preparation methods of HMR-PVDF and HTR-PVDF hollow fiber membranes should be shown briefly.
Response: It is really true as Reviewer suggested that preparation methods of HMR-PVDF and HTR-PVDF hollow fiber membranes should be shown briefly. In this study, the HMR-PVDF and HTR-PVDF hollow fiber membranes were prepared by the concentric spinning technology. The supplementary content and references were listed as following:
In this work, we prepared HMR-PVDF and HTR-PVDF hollow fiber membranes through the concentric spinning technology which consisted of separation and supporting layer. The schematic diagram of preparation process and structure of membrane were shown in Fig.1. Firstly, the casting solution could be mixed under constant mechanical stirring in a three-necked round bottom flask for 3 h at 80℃. And then the nascent HMR-PVDF and HTR-PVDF hollow fiber membranes were fabricated by coating the doped-GO solutions on the outer surface of hollow tubular braids. During the spinning process, the coagulation bath was adjusted to 40℃ in order to speed up the exchange of solvents and non-solvents, which was benefit for thermodynamically compatible (producing favorable interfacial bonding layer). Finally, the HMR-PVDF and HTR-PVDF hollow fiber membranes were obtained by storing in water (25℃) for at least 48 h to extract the residual solvents and water-soluble additives. The spinning parameters of HBR CA hollow fiber membranes were shown in Table 2.
Fig.1 Schematic diagram of the spinning process by concentric spinning technology
Table 1. The parameters of the GO-doped HBR CA hollow fiber membranes
Conditions |
Value |
Spinning temperature (℃) |
23 ± 0.1 |
Coagulation bath |
Water |
Coagulation temperature (℃) |
40 ± 3 |
Air gap (cm) |
8 ± 0.5 |
Take-up speed (m h-1). |
0.01 |
- Comment: Line 61-67 should be moved to “Results and discussion” part. The corresponding determining methods of items in Table 1 should be added in Experimental part. Chemicals used in this study should be presented in this part as well.
Response: It is really true as Reviewer suggested that Line 61-67 should be moved to “Results and discussion” part. The corresponding determining methods of items in Table 1 should be added in Experimental part, the properties of the HMR-PVDF and HTR-PVDF membranes were moved to section 3.3.1, and the revised details were shown as following:
2.1. Materials
The chemicals used in this study were supplied by Tianjin Kermel Chemical Rea-gent Co., Ltd. (Tianjin, China). Bovine serum albumin (BSA, Mw=68,000) purchased from Sinopharm Chemical Reagent Co., Ltd. (Shanghai, China).
2.2 Membrane characteristic
The optical contact angle meter (DSA130, KRUSS) and rheometer (HAAKE MARS, Thermo Fisher Scientific) were utilized to measure the water contact angle of membrane and viscosity of casting solutions. The maximum pore size and mean pore size distri-bution were tested by bubble point test equipment (3H-2000PB, Beishide, China). The porosity of hollow fiber membrane was regarded as the pores volume in total volume. In generally, the gravimetric method was usually utilized to calculate it. Firstly, the de-ionized water was selected to soak the membrane sample for 24 h. Then blowing system was employed to remove the water in the pores of membrane. And a filter paper was utilized to remove the water attached on the outer membrane surface. Finally, the weight of wet membrane and dry membrane was tested after drying in electric blast drying oven for 12 h at temperature of 40 ℃.
- Comment: Data in Tables were measured for only once? If not, please identify determination times and the data in the table should be indicated as average value ± standard deviation. Unit in table 1 should be double-checked as well.
Response: According to the reviewer’s comments, we have made a Table of indicating as average value ± standard deviation, and the revised details were shown as following:
Tab. 2 Properties of the membranes used in this work
Items |
HTR-PVDF |
HMR-PVDF |
Average Pore Size (μm) |
0.58±0.01 |
0.53±0.02 |
Porosity (%) |
45.67±0.2 |
50.19±0.35 |
Contact Angle (°) |
74.50±0.5 |
72.00±0.6 |
Protein Rejection Rate (%) |
96.03±1.1 |
97.88±0.9 |
Pure Water Flux(L •m-2•h-1) |
269.34±2.2 |
293.82±2.5 |
- Comment: Line 198-201 should be rewritten.
Response: As the reviewer suggested, we have rewritten Line 198-201, and the revised details were as following:
Considering the hydrophobic interactions [16], the HTR-PVDF hollow fiber membranes exhibited good adsorption and deposition of pollutants on the membrane surface, which could form a denser adsorption layer than the HMR-PVDF membranes, the HTR-PVDF hollow fiber membranes. As observed in Fig.3 (a), the HTR-PVDF membranes were smooth for inclining to achieve equilibrium state and formed a thick layer which arose from adsorption of solutes or colloids within/on membranes during the static procedure.
- Comment: Format of Line 203 should be double-checked.
Response: According to the reviewer’s comments, we have rewritten Line 203, and the revised details were as following:
As observed in Fig.3 (a), the HTR-PVDF membranes were smooth for inclining to achieve equilibrium state and formed a thick layer during the static adsorption process. Moreover, the outer surfaces SEM morphology of HMR-PVDF membranes were shown in the Fig.S3.
- Comment: Roughness data (Ra, Rq and Rmax) of HTR-PVDF and HMR-PVDF membranes should be shown in this paper.
Response: It is really true as Reviewer suggested that roughness data (Ra, Rq and Rmax) of HTR-PVDF and HMR-PVDF membranes should be shown in this paper. The roughness of the membranes were observed by confocal laser scanning microscope (CSM700, Zeiss) and the revised details were as following:
Roughness of membrane surface played a key role on the particles adhesion as well. It indicated that a loose fouling layer was produced in the rough membranes (HMR-PVDF) resulting in a low flow resistance (per unit thickness of foulant) rather than the dense fouling layers (HTR-PVDF).
Fig.S1 The roughness of HTR-PVDF and HMR-PVDF membranes, before and after fouling
- Comment: Format in this paper should be double-checked, such as “Table”/”Tab.”
Response: Considering the Reviewer’s suggestion, we have revised the full paper carefully and tried to make format reasonable and acceptable. For example, The feed water was prepared by mimicking industrial waste water, the water quality was presented in Tab 1.

Reviewer 2 Report
The main aim of the present work was to study the fouling properties of HMR-PVDF and HTR-PVDF hollow fiber membranes.
The work is interesting. The manuscript is well organized, the research design is appropriate and the methods are adequately described. English is fine.
However, before the publication, the manuscript should be improved. Please, see the comments below:
1. Certainly, the Introduction must be improved. Indeed, it does not provide sufficient background and it does not include all relevant references:
- The first paragraph: The Authors discuss the MBR. The MBR is “the best available technology" for the wastewater treatment, hence its advantages should be clearly presented and emphasized. Please see recently published papers focused on this topic, for instance:
DOI: 10.1016/j.scitotenv.2021.152132
DOI: 10.3390/en15144981
DOI: 10.1016/j.biortech.2021.125793
- The second paragraph: The Authors discuss the use of polymeric membranes. The adevntages of polymeric membranes should be better discussed. The justification for the use of polymeric membranes in this work should be emphasized. The comparison of polymeric and ceramic membranes should be presented. Please, see recently published papers focused on this topic, for instance:
DOI: 10.1016/j.memsci.2020.118987
DOI: 10.1007/s40201-021-00784-w
DOI: 10.3390/membranes11010044
- The third paragraph: The Authors discuss the fouling. It is very important phenomenon and it should be better discussed. The consequences of the phenomenon should be discussed and ways of reducing it (membranes cleaning etc.) should be presented. There is no reference to many works on this issue. You can find it, for instance, in the following papers:
- DOI: 10.3390/membranes10040067
- DOI: 10.1016/j.watres.2022.118269
- The fourth paragraph: the Authors present the aim of the work. What is the novelty of this work? It should be pointed out.
- Although the presented topic is widely presented in the literature, the Authors cited only 23 works. Hence, a more in-depth review of the literature is required.
2. Section Results and discussions should be improved. Indeed, the obtained results should be compared with those available in the literature. It is due to the fact that the models used by the Authors were used earlier in many works.
Author Response
Reviewer: #2
The main aim of the present work was to study the fouling properties of HMR-PVDF and HTR-PVDF hollow fiber membranes.
The work is interesting. The manuscript is well organized, the research design is appropriate and the methods are adequately described. English is fine.
However, before the publication, the manuscript should be improved. Please, see the comments below:
Technical Aspects:
- Comment: Certainly, the Introduction must be improved. Indeed, it does not provide sufficient background and it does not include all relevant references:
The first paragraph: The Authors discuss the MBR. The MBR is “the best available technology" for the wastewater treatment, hence its advantages should be clearly presented and emphasized. Please see recently published papers focused on this topic, for instance:
DOI: 10.1016/j.scitotenv.2021.152132
DOI: 10.3390/en15144981
DOI: 10.1016/j.biortech.2021.125793
The second paragraph: The Authors discuss the use of polymeric membranes. The adevntages of polymeric membranes should be better discussed. The justification for the use of polymeric membranes in this work should be emphasized. The comparison of polymeric and ceramic membranes should be presented. Please, see recently published papers focused on this topic, for instance:
DOI: 10.1016/j.memsci.2020.118987
DOI: 10.1007/s40201-021-00784-w
DOI: 10.3390/membranes11010044
The third paragraph: The Authors discuss the fouling. It is very important phenomenon and it should be better discussed. The consequences of the phenomenon should be discussed and ways of reducing it (membranes cleaning etc.) should be presented. There is no reference to many works on this issue. You can find it, for instance, in the following papers:
- DOI: 10.3390/membranes10040067
- DOI: 10.1016/j.watres.2022.118269
The fourth paragraph: the Authors present the aim of the work. What is the novelty of this work? It should be pointed out.
- Although the presented topic is widely presented in the literature, the Authors cited only 23 works. Hence, a more in-depth review of the literature is required.
Response: According to the reviewer’s comments, we have stated some of arguments in this paper by citing relevant references. The revised details were shown as following:
It is well-known that the membrane bioreactor (MBR) technology is a combination of the conventional activated sludge (ASP) and membrane separation process, which has been applied to wastewater treatment. As we known, the industrial dairy waste contained a high concentration of organic material such as proteins, which had high levels of chemical oxygen demand (COD), total Kjeldahl nitrogen (TKN), and high concentrations of suspended solids (SS). Through membrane filtration, the wastewater could be reused partially or followed by deep filtration. [1-3]. Especially in MBR system, the hollow fiber membrane was prone to get affected by long-term water flow pulsa-tion or disturbance, which was easily damaged and destroyed by high-pressure hy-draulic cleaning process or high-speed water flow disturbance or even aeration process. Up to now, the enhanced mechanical properties of hollow fiber membranes could meet the actual requirement by the reinforced method[4-6]. In general, the reinforced hollow fiber membranes were usually constructed by introducing the braid tube. Moreover, the homogenous reinforced (HMR) and heterogeneous-reinforced (HTR) hollow fiber membranes were put forward through the same or different separation and supporting layer[7-9].
Nowadays, the polyvinylidene fluoride (PVDF) hollow fiber membranes are widely used in MBR system. Many works had been reported by the improvement of antifouling of PVDF hollow fiber membrane. For example, Hikita[10] et al. studied amphiphilic graft copolymers as coatings on the commercially available PVDF membranes. Although the composite membranes possessed good antifouling property, the separation (coating) layer was easy to be peeled off. Liu [11] et al. also designed the antifouling PVDF mem-brane by blending TiO2. It could be found that this method play an important in anti-fouling of outer surface. However, the big pore structure was generated with pore blocking, which might develop permanent fouling for PVDF hollow fiber membrane. So far, the reinforced hollow fiber membrane including heterogeneous reinforced and homogeneous reinforced was suitable way to overcome the limitations of poor tensile strength. At beginning, the heterogeneous reinforced was first proposed by coating the polymer solutions on the outer surface of high-strength hollow tubular braid. However, thermodynamically incompatible between the reinforced fiber and porous membrane caused the separation layer easily peeling from the tubular braid. Comparing with aforementioned heterogeneous reinforced method, the advantages of homogeneous reinforced that contained the same materials in the separation layer and the supporting layer exhibited thermodynamically compatible with good interfacial bonding strength. The results indicated the favorable interfacial bonding between the separation layer and the high-strength hollow tubular braid, avoiding the peeling. However, the favorable interfacial bonding could produce the dense interface, which also affected the permea-bility. [12-14]
Based on some previous works, the fouling mechanism of membrane separation process including the reversible and irreversible fouling or the formation of cake and gel layer was also widely researched[15-16]. On the one hand, the understand of fouling mechanism could help to optimize membrane structure and properties. On the other hand, it is significant for prolong the service life of membrane during the MBR process. For instance, Du[17] found that one of the vital factor was the hydrophilic and hydro-phobic properties of membrane, which could result in the faster irreversible fouling with more hydrophobic. Consequently, the fouling mechanism and antifouling prop-erties of reinforced PVDF hollow fiber membrane will exhibit promising research sig-nificance for a wide range of membrane applications in MBR system.
[1] Sengar A, Vijayanandan A. Effects of pharmaceuticals on membrane bioreactor: Review on membrane fouling mechanisms and fouling control strategies[J]. Science of The Total Environment, 2022, 808: 152132.
[2] Tomczak W, Gryta M. Energy-Efficient AnMBRs Technology for Treatment of Wastewaters: A Review[J]. Energies, 2022, 15(14): 4981.
[3] Akkoyunlu B, Daly S, Casey E. Membrane bioreactors for the production of value-added products: Recent developments, challenges and perspectives[J]. Bioresource Technology, 2021, 341: 125793.
[12] Yadav P, Ismail N, Essalhi M, et al. Assessment of the environmental impact of polymeric membrane production[J]. Journal of Membrane Science, 2021, 622: 118987.
[13] Hashemi T, Mehrnia M R, Ghezelgheshlaghi S. Influence of alumina nanoparticles on the performance of polyacrylonitrile membranes in MBR[J]. Journal of Environmental Health Science and Engineering, 2022, 20(1): 375-384.
[14] Tomczak W, Gryta M. Comparison of polypropylene and ceramic microfiltration membranes applied for separation of 1, 3-PD fermentation broths and Saccharomyces cerevisiae yeast suspensions[J]. Membranes, 2021, 11(1): 44.
[15] Tomczak W, Gryta M. Cross-flow microfiltration of glycerol fermentation broths with Citrobacter freundii[J]. Membranes, 2020, 10(4): 67.
[16] Jarvis P, Carra I, Jafari M, et al. Ceramic vs polymeric membrane implementation for potable water treatment[J]. Water Research, 2022, 215: 118269.
- Comment: Section Results and discussions should be improved. Indeed, the obtained results should be compared with those available in the literature. It is due to the fact that the models used by the Authors were used earlier in many works.
Response: It is really true as Reviewer suggested that section results and discussions should be improved. We have revised the full paper carefully and tried to reframe the sentences. For example, the revised contents from Line 213-219: Considering the hydrophobic interactions [24], the HTR-PVDF hollow fiber membranes exhibited good adsorption and deposition of pollutants on the membrane surface, which could form a denser adsorption layer than the HMR-PVDF membranes, the HTR-PVDF hollow fiber membranes. As observed in Fig.3 (a), the HTR-PVDF membranes were smooth for inclining to achieve equilibrium state and formed a thick layer during the static adsorption process. Moreover, the outer surfaces SEM morphology of HMR-PVDF membranes were shown in the Fig.S3.
Finally, special thanks to you for your good comments.
We have tried our best to improve the manuscript and made some changes in the manuscript. We appreciate for your warm work earnestly and hope that the correction will meet with approval. Thanks a lot for your comments and suggestions.

Round 2
Reviewer 1 Report
The modified manuscript meets the criteria for publication.
Reviewer 2 Report
The manuscript has been improved, hence, I recommend it for publication in present form.
Congratulations.